# ADAPTIVE MULTILEVEL NEURAL NETWORKS FOR PARAMETRIC PDEs WITH ERROR ESTIMATION

**Janina E. Schütte, Martin Eigel**
Weierstraß-Institut für Angewandte Analysis und Stochastik (WIAS)
Mohrenstr. 39, 10117 Berlin, Germany
`{schuette, eigel}@wias-berlin.de`

## ABSTRACT

To solve high-dimensional parameter-dependent partial differential equations (pPDEs), a neural network architecture is presented. It is constructed to map parameters of the model data to corresponding finite element solutions. To improve training efficiency and to enable control of the approximation error, the network mimics an adaptive finite element method (AFEM). It outputs a coarse grid solution and a series of corrections as produced in an AFEM, allowing a tracking of the error decay over successive layers of the network. The observed errors are measured by a reliable residual based a posteriori error estimator, enabling the reduction to only few parameters for the approximation in the output of the network. This leads to a problem adapted representation of the solution on locally refined grids. Furthermore, each solution of the AFEM is discretized in a hierarchical basis. For the architecture, convolutional neural networks (CNNs) are chosen. The hierarchical basis then allows to handle sparse images for finely discretized meshes. Additionally, as corrections on finer levels decrease in amplitude, i.e., importance for the overall approximation, the accuracy of the network approximation is allowed to decrease successively. This can either be incorporated in the number of generated high fidelity samples used for training or the size of the network components responsible for the fine grid outputs. The architecture is described and preliminary numerical examples are presented.

## 1 INTRODUCTION

The ability to quickly solve pPDEs is crucial for addressing real-world problems with dynamic, uncertain and changing conditions, enabling the exploration of a wide range of scenarios. They arise in diverse fields such as engineering, environmental science, and finance. Problems of this type have been examined extensively in Uncertainty Quantification (UQ) in recent years, see e.g. Lord et al. (2014). In the realm of computational fluid dynamics, solving *parametric stationary diffusion* (Darcy flow) problems efficiently across a spectrum of parameters remains an important challenge. For a possibly countably infinite dimensional parameter space $\Gamma \subseteq \mathbb{R}^{\mathbb{N}}$ and a physical domain $D \subseteq \mathbb{R}^d$, $d \in \{1, 2\}$, the objective is to find a solution $u : D \times \Gamma \to \mathbb{R}$ such that

$$
\begin{aligned}
-\nabla \cdot (\kappa(\cdot, \mathbf{y}) \nabla u(\cdot, \mathbf{y})) &= f && \text{on } D, \\
u(\cdot, \mathbf{y}) &= 0 && \text{on } \partial D
\end{aligned}
\tag{1.1}
$$

holds for every $\mathbf{y} \in \Gamma$ with a permeability parameter field $\kappa : D \times \Gamma \to \mathbb{R}$. For each parameter $\mathbf{y} \in \Gamma$ the solution $u(\cdot, \mathbf{y})$ can be approximated point wise, e.g. with a finite element solver. To calculate a quantity of interest, Monte Carlo methods can be used to explore the parameter domain $\Gamma$. Since this technique is computationally expensive, more advanced numerical methods for the computation of functional surrogates such as stochastic collocation Babuška et al. (2007); Teckentrup et al. (2015); Nobile et al. (2008) or low-rank tensor approximations of adaptive stochastic Galerkin formulations Eigel et al. (2020; 2014) have been examined, by which the statistical properties of the parameter become directly accessible. Neural network surrogate models such as DeepONet in an infinite dimensional setting have been analyzed in Chen and Chen (1995); Lanthaler et al. (2022); Lu et al. (2021); Wang et al. (2021); Marcati and Schwab (2023), Fourier neural operators were introduced in li2 (2021) and in a discretized setting the problem is combined with reduced basis methods in Kytyniok et al. (2022); Geist et al. (2021); Dal Santo et al. (2020). The expressivity results derived in these works mostly consider fully connected neural networks, which are based on the seminal work Yarotsky (2017) for polynomial approximation including dependence on the parameter dimension. In He and Xu (2019) and

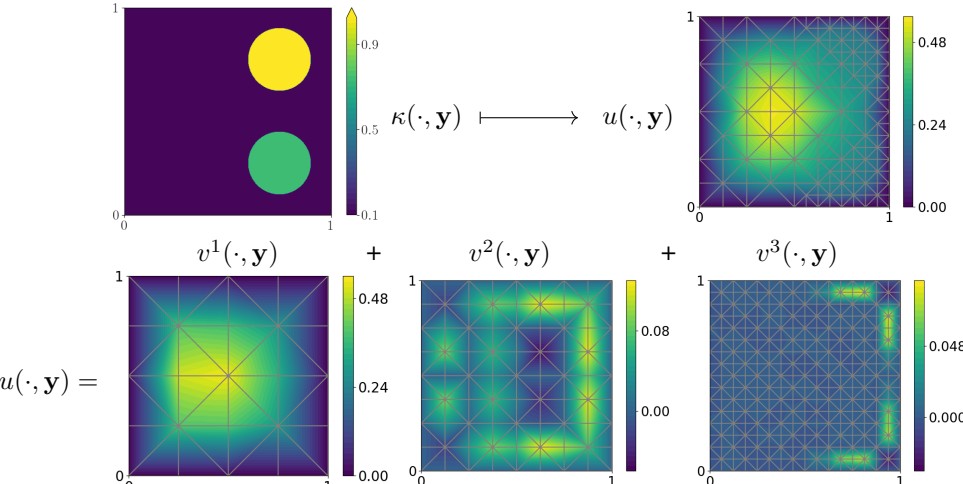

Figure 1.1: The first row depicts the parameter $\kappa$ to solution $u$ map for a realization of $\mathbf{y} \in \Gamma$ for the parametric stationary diffusion PDE. In the second row, the applied multigrid decomposition of the solutions into a coarse grid function $v^1$ and finer grid corrections $v^2, v^3$ is visualized.

Chen et al. (2022), connections between CNNs and multigrid methods to solve PDEs were studied. In Heiß et al. (2023), a convolutional CNN was introduced and analyzed, mapping parameters of a PDE model to discrete solutions. The expressivity results alleviate the dependence on the parameter dimension and numerical examples illustrate SOTA results. In Caboussat et al. (2024), adaptively created meshes are used to train a fully connected neural network, mapping the parameter and point in the physical domain to the evaluation of the corresponding solution.

In this context, our paper introduces an approach based on CNNs to quickly solve the Darcy flow problem using a finite element (FE) discretization on locally refined grids based on an adaptive FE scheme and an a posteriori error estimator, in contrast to uniformly refined grids as in Heiß et al. (2023). The network maps the FE discretized parameter field $\kappa(\cdot, \mathbf{y})$ to the solution $u(\cdot, \mathbf{y})$ on a locally refined grid for parameters $\mathbf{y} \in \Gamma$, see the first row in Figure 1.1. The derived architecture is inspired by an AFEM with a successive subspace correction algorithm as an algebraic solver on a fixed grid. It iterates the steps

$$\text{Solve} \;\rightarrow\; \text{Estimate} \;\rightarrow\; \text{Mark} \;\rightarrow\; \text{Refine}$$

based on a reliable and efficient local error estimator, see Appendix B. To mark elements to be refined for the next iteration, Dörfler marking or threshold marking can be used. A newest vertex bisection method is implemented for mesh refinement, see Appendix C.

Additionally, a multi-level discretization of the solutions is incorporated into the derived CNN architecture, see the second row in Figure 1.1. This multi-level decomposition leads to a more efficient approach in the number of required samples (in particular for the finer levels), the training process and the overall size of the network, see Heiß et al. (2023); Lye et al. (2020) for details. The proposed architecture is introduced in Section 2.2. Numerical tests are provided for discretizations on fixed locally refined grids in Section 3.

## 2 ADAPTIVE FEM INSPIRED CNN ARCHITECTURE

CNNs are a special class of neural networks designed for tasks involving image data such as image classification and segmentation. Applying the action of a CNN to an image involves the application of local kernels to a hierarchy of representations of the input. This locality makes the architecture especially useful in partial differential equations, where local properties have to be resolved to obtain highly accurate representations. To incorporate interactions on a larger scale with respect to the image domain, successive down-scaling of the input images can be implemented with CNNs through strided and transpose strided convolutions leading to the popular CNN architecture of Unets Ronneberger et al. (2015). This architecture is heavily exploited in Heiß et al. (2023), where it is shown that Unets can approximate

multigrid solvers. In the analysis of the implemented CNN architecture, the multiresolution images are used as the representation of the solutions of the parametric PDE with respect to coarse and fine grid FEM discretizations.

**Theorem 2.1** ((Heiß et al., 2023, Corollary 8)). *Let $\mathbf{u_y}$ be the FE coefficients of the Galerkin projection of the solution of Equation* (1.1) *onto the piecewise affine FE space over a uniform square mesh with triangle size $h$. Assume that the parameter fields are uniformly bounded over the parameters. Then for any $\varepsilon > 0$ there exists a CNN $\Psi$ with the discretized parameter field and right-hand side as an input and FE coefficients as an output approximating $\mathbf{u_y}$ with $\varepsilon \|f\|_*$ accuracy for all parameters and the sum over the kernel sizes has complexity $\mathcal{O}(\log(1/h)\log(1/\varepsilon))$.*

To train the derived architecture, pairs of parameter field realizations and solutions of the variational formulation of the Darcy flow equation A.1 are generated by solving the parametric PDE on a uniform grid using the `FEniCS` FE solver Logg et al. (2012). For high resolution solutions, fine grids need to be considered. The number of finite element coefficients grows exponentially with the refinement of the grids for higher resolutions, i.e., lower discretization errors. In their numerical experiments, it was observed that the approximation error of the neural network approximating the finite element coefficients on a fixed uniform grid is magnitudes smaller than the discretization error measuring the distance of the Galerkin projection of the full solution on the grid to the full solution. As computations on finer uniform grids get infeasible quickly, the consideration of locally refined grids has the advantage that only important coefficients with a lot of information are considered. By this, a high accuracy can be achieved with a small number of coefficients. An adaptive method to derive efficient locally refined grids is described in Section 2.1. To incorporate locally refined grids in a CNN architecture, masks localizing the corrections to relevant coefficients can be used. They define the area of an input image, which should be used for further computations, i.e., where the convolutional kernels and activation function should be applied. The architecture is described in Section 2.2.

## 2.1 ADAPTIVE METHOD

To find a grid adapted to the problem, i.e., a coefficient-efficient representation of the solution, for a fixed parameter $\mathbf{y}$ an AFEM can be used. There, a grid is constructed iteratively. Starting with the solution on a coarse grid as the grid depicted on the bottom left of Figure 1.1, the local error contributions of the following efficient and reliable error estimator is used to identify triangles with a high error, cf. Verfürth (2013); Carstensen et al. (2012) and Eigel et al. (2014) for the parametric setting: For a triangle $T$ in the triangulation with maximal side length $h_T$ the estimator of the form

$$\eta_T^2 := h_T^2 \|f + \nabla \cdot (\kappa(\cdot, \mathbf{y})\nabla u)\|_{L_2(T)}^2 + h_T \|[\![\kappa(\cdot, \mathbf{y})\nabla u]\!]\|_{L_2(\partial T)}^2 \tag{2.1}$$

bounds the error from above in the energy norm, see Appendix B. These triangles are marked either based on a threshold with a fixed allowed error on each triangle or with a Dörfler marking strategy always identifying the largest errors that are left. The refinement of these triangles is done in a way that the resulting meshes can be interpreted as sub-meshes of uniformly refined meshes to get a regular grid needed for the CNN, which applies the same kernel to every pixel/node in the input images, see Appendix C. In the next iterations, the correction of the coarser grid solution on a finer grid and their estimators are computed and a locally refined grid is generated. The advantage of the algorithm compared to uniform grids is visualized in Figure D.1.

The algorithm produces a coarse grid solution $u_{(1)}$ and a series of corrections on increasingly finer grids $v_{(n)}$. Each fine grid correction can be discretized in a hierarchical basis, i.e., in coefficients on the first grid $\mathbf{v}_{(n)}^1$ and alterations on finer grids $\mathbf{v}_{(n)}^i$ (multigrid discretization). With the chosen refinement, each alteration can be interpreted as a function on a uniformly refined mesh with zero coefficients on unrefined triangles and the coefficients can be encoded by masked sparse images.

## 2.2 CNN ARCHITECTURE

We derive an architecture based on the AFEM described above, where each step of the algorithm should be approximated by a specific part of the proposed CNN $\Psi$. The architecture is depicted in Figure 2.1 for two steps of the AFEM and therefore three grids. The input images $\kappa_{\mathbf{y}}, \mathbf{f}$ contain the finite element coefficients of the diffusion coefficient $\kappa(\cdot, \mathbf{y})$ and the right-hand side $f$ discretized on the finest uniformly refined grid. The architecture can be interpreted as follows: In the yellow part of the network, the input can get downsampled to coarser grids. The green, blue and purple Unet-like parts of the network are solvers on the build grids first on a coarse (green) and then finer grids. The brown outputs of the network are the finite element coefficients of the multigrid discretized solution and corrections. The red outputs are the estimator coefficients in each step and the white and blue images are the derived markers based

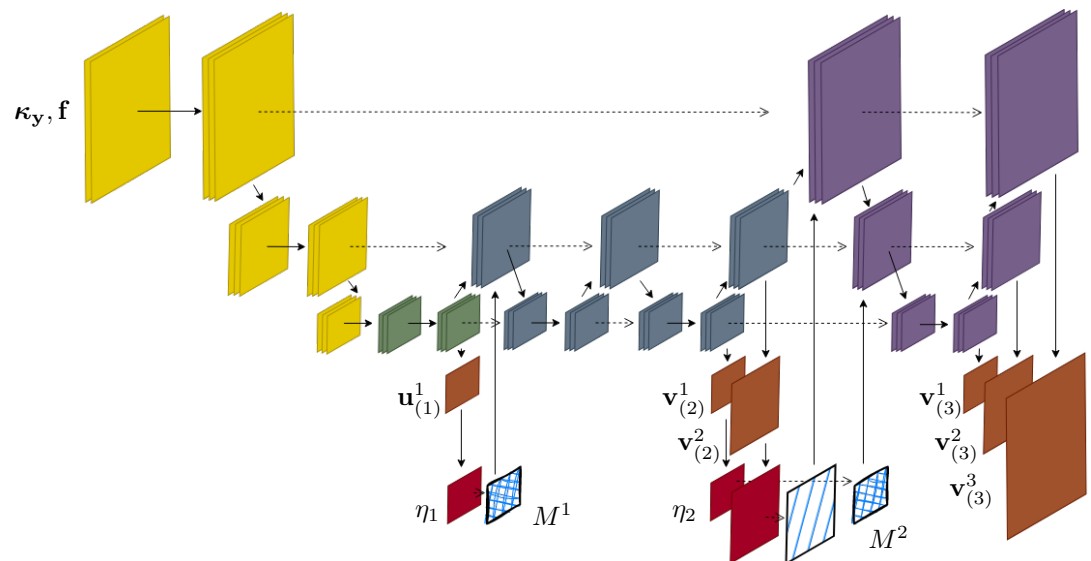

Figure 2.1: The CNN architecture for three levels is depicted.

Table 1: Average relative errors are shown as described in Section 3.

| $p$ | $\mathcal{E}_{\text{NN}}$ | $\mathcal{E}_{\text{total}}$ | $\mathcal{E}_{\text{discr}}$ |
|---|---|---|---|
| $H_0^1$ | $2.82 \times 10^{-3}$ | $2.6357 \times 10^{-1}$ | $2.6354 \times 10^{-1}$ |
| $L^2$ | $1.28 \times 10^{-3}$ | $8.999 \times 10^{-2}$ | $8.991 \times 10^{-2}$ |

on the estimators, which can be calculated inside the network for specific markings or outside of the network for any marking. The $0 - 1$-marking images are then fed back into the network by multiplying the masks to the input images of the next solver.

**Remark 2.1.** *If local marking strategies like thresholding are used, the marking could be learned by the CNN. In case global marking strategies, such as the Dörfler marking, are preferred, the marking can be calculated outside the CNN based on the estimator output of the CNN.*

## 3  NUMERICS

The architecture is realized for three steps of the AFEM on a fixed locally refined grid. The presented results are concerned with the cookie problem with fixed radii and centers with two active cookies, Heiß et al. (2023).

**Definition 3.1** (Cookie problem with $2$ inclusions). *Let $\mathbf{y} \in \Gamma = [0,1]^2$ be uniformly distributed $\mathbf{y} \sim U([0,1]^2)$. Define the parameter field for $x \in D = [0,1]^2$ by*

$$\kappa(x, \mathbf{y}) := 0.1 + \sum_{k=1}^{2} \mathbf{y}_k \chi_{D_k}(x),$$

*where $D_k$ are disks centered at the right side in a $2 \times 2$ cubic lattice with fixed radius $r = 0.15$ for $k = 1, 2$.*

An example parameter field is depicted in Figure 1.1. It is implemented and trained with `pytorch lightning` Falcon and The PyTorch Lightning team (2019) with $10,000$ training samples generated with a `FEniCS` solver. The architecture was chosen to have $3, 2, 1$ Unet-like structures to approximate the first, second and third step of the AFEM, respectively. This lead to a total number of $1937804$ trainable parameters. In Table 1 average relative errors of the

method are shown. The following errors were calculated for $p \in \{H_0^1, L^2\}$ and for a set of 100 test parameters $\mathcal{Y} \subset \Gamma$:

$$\mathcal{E}_{\text{NN}} := \sum_{\mathbf{y} \in \mathcal{Y}} \frac{\|\Psi(\boldsymbol{\kappa}_{\mathbf{y}}, \mathbf{f}) - \mathbf{u}_{\mathbf{y}}\|_p}{|\mathcal{Y}| \|\mathbf{u}_{\mathbf{y}}\|_p}, \quad \mathcal{E}_{\text{total}} := \sum_{\mathbf{y} \in \mathcal{Y}} \frac{\|\mathcal{C}(\Psi(\boldsymbol{\kappa}_{\mathbf{y}}, \mathbf{f})) - u(\cdot, \mathbf{y})\|_p}{|\mathcal{Y}| \|u(\cdot, \mathbf{y})\|_p} \quad \mathcal{E}_{\text{discr}} := \sum_{\mathbf{y} \in \mathcal{Y}} \frac{\|\mathcal{C}(\mathbf{u}_{\mathbf{y}}) - u(\cdot, \mathbf{y})\|_p}{|\mathcal{Y}| \|u(\cdot, \mathbf{y})\|_p},$$

the errors of the network to the Galerkin solution on the same grid, the errors of the network to the true solution and the errors of the Galerkin solution to the true solution are depicted, where $\mathcal{C}$ maps the FE coefficients to the corresponding function. The true solution corresponds to a Galerkin solution on a twice uniformly refined grid. It can be observed that the local corrections can be learned and the error of the network is insignificant in comparison to the discretization error. Since our adaptive architecture was devised to decrease the discretization error observed in Heiß et al. (2023), the scalability of this approach has to be tested in upcoming research.

## 4  OUTLOOK

The proposed advantage over existing methods on uniformly refined grids to be able to achieve lower discretization errors should be tested for high resolution solutions. To this end, the full proposed architecture should be tested with more AFEM steps. Furthermore, the network should further be investigated on adaptive grids as proposed in the derivation of the architecture. For this purpose, the estimator should be approximated as well. Furthermore, the architecture should be analyzed and expressivity results should be derived, which is current ongoing work. The introduced method is restricted to PDEs, which allow the derivation of a suitable error estimator leading to an efficient approximation of solutions with the AFEM.

ACKNOWLEDGMENTS

We acknowledge funding by the DFG SPP 2298 - Theoretical Foundations of Deep Learning. The authors want to thank Cosmas Heiß for fruitful discussions.

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

## A  FINITE ELEMENT PRIOR

We assume a regular conforming triangulation $\mathcal{T}$ of the (smoothly bounded) domain $D$. Let $V_h = \operatorname{span}\{\varphi_j\}_{j=1}^{\dim V_h} \subset H_0^1(D)$ be a finite dimensional subspace spanned by conforming first order (Lagrange) basis functions. We consider the discretized solution $u_h \in V_h$ and parameter fields $\kappa_h \in V_h$

$$u_h = \sum_{i=1}^{\dim V_h} \mathbf{u}_i \varphi_i, \quad \kappa_h(\cdot, \mathbf{y}) = \sum_{i=1}^{\dim V_h} (\boldsymbol{\kappa_y})_i \varphi_i,$$

where coefficient vectors with respect to the basis of $V_h$ are written bold face. With slight abuse of notation, let $f \in H^{-1}(D)$. In the variational formulation of Equation (1.1), we are concerned with finding a solution $u_h \in V_h$ for any $\mathbf{y} \in \Gamma$ such that for all $w_h \in V_h$

$$a_{\mathbf{y},h}(u_h, w_h) := \int_D \kappa_h(\cdot, \mathbf{y}) \langle \nabla u_h, \nabla w_h \rangle \mathrm{d}x = \int_D f w_h \mathrm{d}x =: f(w_h). \tag{A.1}$$

This is equivalent to determining $\mathbf{u} \in \mathbb{R}^{\dim V_h}$ by solving the algebraic system

$$A_{\mathbf{y}} \mathbf{u} = \mathbf{f}$$

with the right-hand side $\mathbf{f} := (f(\varphi_j))_{j=1}^{\dim V_h}$ and discretized operator $A_{\mathbf{y}} := (a_{\mathbf{y},h}(\varphi_i, \varphi_j))_{i,j=1}^{\dim V_h}$.

We consider the norms for $T \subset \mathbb{R}^d$ and $u : \mathbb{R}^d \to \mathbb{R}$ given by

$$\|u\|_{a_{\mathbf{y},h}}^2 := a_{\mathbf{y},h}(u, u), \qquad \|u\|_{L^2(T)}^2 := \int_T u^2 \mathrm{d}x.$$

## B  ERROR ESTIMATION

We recollect the common residual based a posteriori error estimator for the Galerkin solution $u_h$ of the (parametric) Darcy problem in $V_h$ solving Equation (A.1).

**Definition B.1** (Jump & error estimator). *We define the* jump *along the edge between triangles $T^1$ and $T^2$ with $\nabla u^{(1)}$ and $\nabla u^{(2)}$ the gradients on the triangles, respectively, by*

$$[\![\kappa_h(\cdot, \mathbf{y}) \nabla u_h]\!] := \kappa_h(\cdot, \mathbf{y}) \left( \left\langle \nabla u_h^{(1)}, n_\gamma^{(1)} \right\rangle + \left\langle \nabla u_h^{(2)}, n_\gamma^{(2)} \right\rangle \right). \tag{B.1}$$

*We set the local error contribution on each triangle $T$ to*

$$\eta_T^2 := h_T^2 \|f + \nabla \cdot (\kappa_h(\cdot, \mathbf{y}) \nabla u_h)\|_{L_2(T)}^2 + h_T \|[\![\kappa_h(\cdot, \mathbf{y}) \nabla u_h]\!]\|_{L_2(\partial T)}^2.$$

The derivation of the reliability of the estimator

$$\|u - u_h\|_{a_{\mathbf{y},h}}^2 \leq C \sum_{T \in \mathcal{T}} \eta_T^2.$$

can be found in Appendix B.1.

### B.1  ERROR ESTIMATOR DERIVATION

Consider the residual in variational form for the error $e := u - u_h$, where $u_h$ is the Galerkin approximation of $u$ on $V_h$ and $\mathcal{T}$ is the considered triangulation. It then holds that

$$a_{\mathbf{y},h}(e, v) = a_{\mathbf{y},h}(u, v) - a_{\mathbf{y},h}(u_h, v) = f(v) - a_{\mathbf{y},h}(u_h, v) = \int_D fv - \kappa_h(\cdot, \mathbf{y}) \langle \nabla u_h, \nabla v \rangle \mathrm{d}x$$

$$= \sum_{T \in \mathcal{T}} \int_T fv - \kappa_h(\cdot, \mathbf{y}) \langle \nabla u_h, \nabla v \rangle \mathrm{d}x.$$

Furthermore, let $n_T$ be the unit outward normal vector to $\partial T$ for $T \in \mathcal{T}$. Then, it holds that

$$a_{\mathbf{y},h}(e,v) = \sum_{T \in \mathcal{T}} \int_T fv\mathrm{d}x + \int_T v\nabla \cdot (\kappa_h(\cdot,\mathbf{y})\nabla u_h)\mathrm{d}x - \int_{\partial T} v\kappa_h(\cdot,\mathbf{y})\frac{\partial u_h}{\partial n_T}\mathrm{d}s$$

$$= \sum_{T \in \mathcal{T}} \int_T (f + \nabla \cdot (\kappa_h(\cdot,\mathbf{y})\nabla u_h))v\mathrm{d}x + \sum_{\gamma \in \partial \mathcal{T}} \int_\gamma v\kappa_h(\cdot,\mathbf{y})\left(\left\langle \nabla u_h^{(1)}, n_\gamma^{(1)} \right\rangle + \left\langle \nabla u_h^{(2)}, n_\gamma^{(2)} \right\rangle\right)\mathrm{d}s.$$

Here, $n_\gamma^{(1)}$ and $n_\gamma^{(2)}$ are the unit outward normal vectors of the elements of the mesh containing $\gamma$ and $\nabla u_h^{(1)}, \nabla u_h^{(2)}$ are the gradients of $u_h$ on the elements. Then with $v_h$ the Galerkin projection of $v$ on $V_h$ and the definition of the jump Equation (B.1)

$$a_{\mathbf{y},h}(e,v) = \sum_{T \in \mathcal{T}} \int_T (f + \nabla \cdot (\kappa_h(\cdot,\mathbf{y})\nabla u_h))(v - v_h)\mathrm{d}x + \sum_{\gamma \in \partial \mathcal{T}} \int_\gamma [\![\kappa(\cdot,\mathbf{y})\nabla u_h \cdot \hat{n}]\!](v - v_h)\mathrm{d}s$$

$$\leq \sum_{T \in \mathcal{T}} \|f + \nabla \cdot (\kappa_h(\cdot,\mathbf{y})\nabla u_h\|_{L_2(T)} \|v - v_h\|_{L_2(T)} + \sum_{\gamma \in \partial \mathcal{T}} \|[\![\kappa_h(\cdot,\mathbf{y})\nabla u_h]\!]\|_{L_2(\gamma)} \|v - v_h\|_{L_2(\gamma)}$$

$$\leq \tilde{C}\|v\|_{H^1(\Omega)} \left( \sum_{T \in \mathcal{T}} h_T^2 \|f + \nabla \cdot (\kappa_h(\cdot,\mathbf{y})\nabla u_h)\|_{L_2(T)}^2 + \sum_{\gamma \in \partial \mathcal{T}} h_E \|[\![\kappa_h(\cdot,\mathbf{y})\nabla u_h]\!]\|_{L_2(\gamma)}^2 \right)^{1/2}.$$

Setting $v = e$ and with $\|v\|_{H^1(\Omega)} \leq C\|v\|_{a_{\mathbf{y},h}}$ we arrive at a computable upper bound of the energy error in terms of the previously defined error estimator, namely

$$\|e\|_{a_{\mathbf{y},h}}^2 = \frac{(\|e\|_{a_{\mathbf{y},h}}^2)^2}{\|e\|_{a_{\mathbf{y},h}}^2} = \frac{(a_{\mathbf{y},h}(e,e))^2}{\|e\|_{a_{\mathbf{y},h}}^2}$$

$$\leq \frac{1}{\|e\|_{a_{\mathbf{y},h}}^2}\tilde{C}^2 C^2 \|e\|_{a_{\mathbf{y},h}}^2 \left( \sum_{T \in \mathcal{T}} h_T^2 \|f + \nabla \cdot (\kappa_h(\cdot,\mathbf{y})\nabla u_h)\|_{L_2(T)}^2 + \sum_{\gamma \in \partial \mathcal{T}} h_T \|[\![\kappa_h(\cdot,\mathbf{y})\nabla u_h]\!]\|_{L_2(\gamma)}^2 \right)$$

$$\leq \hat{C} \sum_{T \in \mathcal{T}} h_T^2 \|f + \nabla \cdot (\kappa_h(\cdot,\mathbf{y})\nabla u_h)\|_{L_2(T)}^2 + h_T \|[\![\kappa_h(\cdot,\mathbf{y})\nabla u_h]\!]\|_{L_2(\partial T)}^2.$$

# C  Mesh refinement

To ensure that each function on a locally refined mesh can be expressed as a function on a uniformly refined mesh, we consider the following refinement strategy. For an initial triangulation, we define the longest edge of each triangle to be the refinement edge and set the vertex not in the refinement edge as the initial newest vertex. We start by defining the RefineInPair algorithm.

RefineInPair($T$):

- If the refinement edge of $T$ is also the refinement edge of the adjacent triangle $\tilde{T}$ along the edge, connect the mid point of the refinement edge to the newest vertices of $T$ and $\tilde{T}$.

- Otherwise, choose the $T$ adjacent triangle after applying RefineInPair($\tilde{T}$) and connect the mid point as above.

To refine a triangle $T$, we apply RefineInPair to $T$ and then again to the two new triangles, which are subsets of $T$. The newest vertex bisection ensures that the shape of the triangles does not degenerate and refining the neighbouring triangles takes care of hanging nodes and sets new vertices and edges as they would appear in a uniform mesh refinement. For more details we refer to Mitchell (2016).

# D  Adaptive advantage

The advantage of the adaptive algorithm is that only a small number of FE coefficients need to be recovered in every step, which leads to fast computations as well as a coefficient efficient representation of the solution. In Figure D.1,

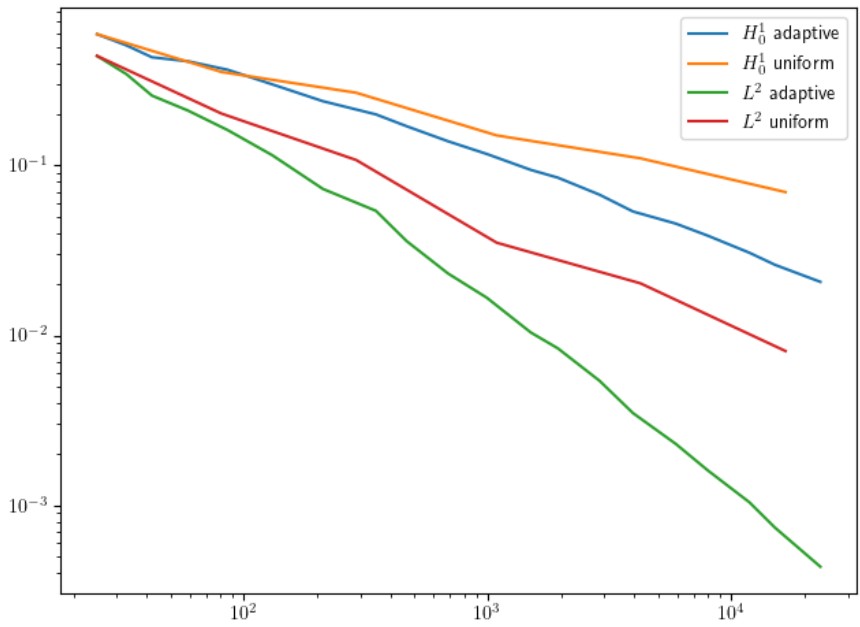

Figure D.1: Plotted are the errors over the number of coefficients of the representation of the solution computed on uniform meshes and on locally refined meshes.

the error decay of the adaptive algorithm is plotted over the number of FE coefficients. As a comparison the error decay over uniformly refined meshes is plotted as well. The error is measured in the $H_0^1$ and the $L^2$ distance of the corresponding finite element functions to a Galerkin projection of the solution onto a function space on a twice uniformly refined mesh. For specific mesh refinements and marking strategies, constant error deacy can be shown for AFEM, e.g. see Dörfler (1996); Chen et al. (2012); Eigel et al. (2015). The error deacy over the steps of the neural network are shown in Figure D.2 for 3 grids.

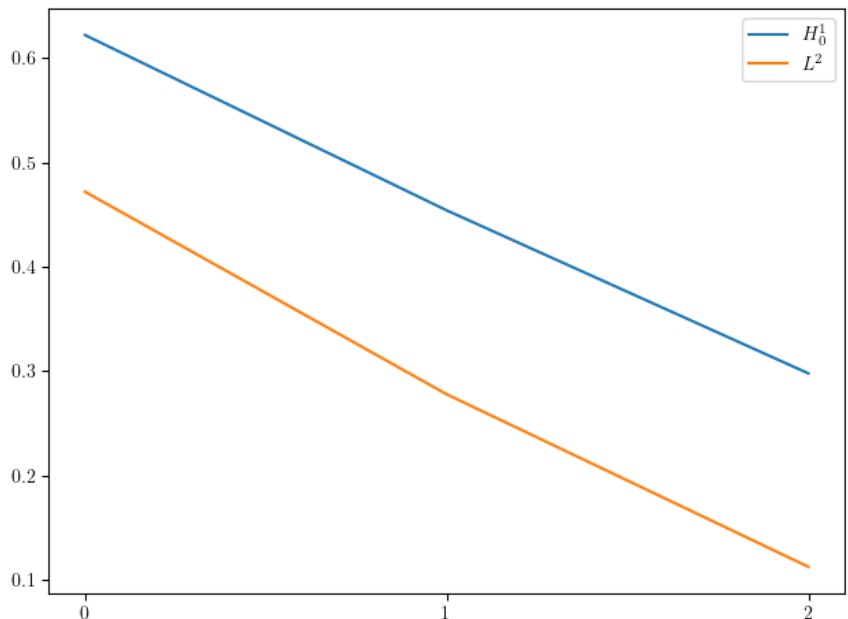

Figure D.2: The decay of the errors over the steps of the CNN are shown.

