# OpenReview forum: "Adaptive Multilevel Neural Networks for Parametric PDEs with Error Estimation"
_ICLR.cc/2024/Workshop/AI4DiffEqtnsInSci — AI4DiffEqtnsInSci @ ICLR 2024 Poster_

### Official Review · Reviewer_QWH3 · 2024-02-13
**an approach based on CNNs accompanied by finite element discretization**

**Rating:** 8
**Confidence:** 5

**Review:**

In the present paper, based on the authors’ claim, an approach based on CNNs accompanied by finite element discretization has been constructed to solve the Darcy flow problem. Besides, a multi-level discretization of the solutions is incorporated into the derived CNN architecture, leading to a more efficient approach in the number of required samples, the training process, and the overall size of the network. Finally, Numerical tests are provided for discretization on fixed locally refined grids.
This paper provides adaptive multilayer neural networks for parametric
PDEs, and may appear to have much interest in the ICLR 2024 Workshop AI4DiffEqtnsInSci readership.
However, the authors did not mention some critical and intriguing features of the CNN architecture, the number of parameters, and the specific benefits of the proposed method in comparison with existing methods. Moreover, more explanation about Table 1 can help interested readers to understand better the outcome of the method.
In addition, talking about the limitation of the proposed method is not available in the paper.

---

### Official Review · Reviewer_zDFi · 2024-02-20
**Adaptive Multilevel Neural Networks for Parametric PDEs with Error Estimation**

**Rating:** 6
**Confidence:** 3

**Review:**

This is good work, and the authors must show the error bounds when kappa is a function of solution itself.

---

### Meta-Review · Area_Chair_nH6C · 2024-03-03

**Recommendation:** Accept (Poster)

**Metareview:**

Both reviewers agree on acceptance. I also vote for acceptance and encourage the authors to address the reviewers' comments in the camera-ready version.

---

### Decision · Program_Chairs · 2024-03-03

Accept (Poster)